# Knowledge and Awareness of Obesity-Related Breast Cancer Risk Among Women in the Qassim Region, Saudi Arabia: A Cross-Sectional Study

**DOI:** 10.3390/healthcare13030278

**Published:** 2025-01-30

**Authors:** Amal Mohamad Husein Mackawy, Manal Alharbi, Mohamad Elsayed Hasan Badawy, Hajed Obaid Abdullah Alharbi

**Affiliations:** 1Department of Medical Laboratories, College of Applied Medical Sciences, Qassim University, Almulaida 52571, Saudi Arabia; 2Department of Medical Biochemistry and Molecular Biology, Faculty of Medicine, Zagazig University, Zagazig City 7120730, Egypt; 3Medical Laboratory Specialist, Medical Laboratory, Applied Medical Sciences College, Qassim University, Almulaida 52571, Saudi Arabia; mannal.777@hotmail.com; 4Consultant of Internal Medicine, Maternal and Child Health Hospital, Buraidah 51452, Saudi Arabia; mbadawy27@gmail.com

**Keywords:** breast cancer, risk factors, obesity, knowledge, awareness, breast self-examination, women’s health screening

## Abstract

**Background**: Breast cancer (BC) is a major health concern globally and the second leading cause of cancer-related mortality in women in Saudi Arabia. Although peoples’ awareness of BC risk factors has been previously examined, studies on obesity-related BC awareness in the Qassim region are inconclusive. We aimed to evaluate knowledge and awareness of obesity-related BC risk among Saudi women in the Qassim region. **Methods**: This is a cross-sectional study with a stratified random sampling technique of 400 Saudi women randomly selected from the Qassim region through an online platform and community health centers. An online closed-ended pretested validated structured questionnaire was completed by the participants using a Google Forms link. The categorical variables were frequency and percentage. The chi-square test was used to study the relationship between the dependent and independent variables. **Results**: There is moderate to poor knowledge regarding breast cancer risk factors. The results showed poor knowledge about obesity after menopause as a risk factor for BC (49%). Over half of the participants (51.0%) did not consider obesity a BC risk factor. The need for self-examinations and mammogram screenings showed moderate (59.6%) and poor awareness levels (4.75%). **Conclusions**: The findings highlight a noticeable gap in knowledge and awareness about obesity-related BC risks, as well as a limited awareness of the need for breast self-examinations and mammogram screenings. These results underscore the urgent need for targeted awareness campaigns and educational programs in the Qassim region to address this critical health issue. Promoting breast self-examination practices, weight management, and regular mammogram screenings could significantly enhance early detection, improve prognosis, and reduce BC-related mortality among Saudi women in the Qassim region.

## 1. Introduction

Breast cancer (BC) is a widespread health issue among women and ranks as the most diagnosed cancer in this population [1,2]. BC is currently the fifth leading cause of cancer-related deaths globally and in Saudi Arabia [1,2]. According to the World Health Organization (WHO), the incidence of breast cancer has risen significantly over the last two decades, increasing from an estimated 10 million cases in 2000 to 19.3 million cases in 2020 [3]. Presently, one in five individuals worldwide are expected to face a cancer diagnosis during their lifetime [3]. Projections suggest that this figure will double by 2040. The global death toll from cancer has increased from 6.2 million in 2000 to 10 million in 2020, with one in every six deaths now attributed to cancer [3]. The incidence of BC in Saudi Arabia remains lower than the global average (27.7 vs. 89.9 per 100,000 women per year, respectively) [4]. However, over the past three decades, a marked increase in BC incidence and associated mortality rates has been recorded [5], paralleling changes in BC risk factors [6]. Obesity stands out as a major global health concern, affecting more than 600 million adults globally, which accounts for approximately 13% of the world’s population [7,8]. The prevalence of obesity among Saudi women has risen significantly over the last several decades, increasing from 14.3% in 1975 to 30% in 2001 and 41.2% in 2016. When combined with those who are overweight (BMI ≥ 25 kg/m^2^), the prevalence is approximately 70% [9,10].

Obesity has been identified as linked with BC incidence and is associated with poor cancer prognoses and higher mortality rates [11,12]. Disease characteristics associated with obesity include larger tumors, more positive lymph nodes, higher tumor grades, increased triple-negative disease in premenopausal women, and more hormone-positive tumors in postmenopausal women [13,14]. Furthermore, body mass index (BMI) is linked to variations in BC molecular subtype distribution [15,16].

This association is believed to be related to hormonal systems, insulin, insulin-like growth factors (IGFs), particularly IGF-binding protein-1 (IGF-1), sex hormones, and adipokines [17]. The IGF/insulin system, comprising insulin-like growth factors and insulin, has potent mitogenic and pro-migratory functions, significantly contributing to various cancer types, including BC [18,19]. Within the IGF/insulin signaling network, the insulin-like growth factor receptor (IGF/IR) plays a crucial role in regulating cell metabolism, promoting transformation, and triggering proliferation and metastasis [19].

Obesity is also considered a risk factor for developing BC because of its role in inducing inflammation in adipose tissue and activating macrophages [20]. This process activates pro-inflammatory mediators such as TNF-α and IL-6. Dysregulated inflammation in adipose tissue results in the accumulation of pro-inflammatory T cells and a decrease in T regs, which is relevant to obesity-related insulin resistance [20,21].

Obesity among Saudi women has been linked to BC incidence and is associated with poor prognosis and increased mortality rates [10]. The level of BC awareness can be measured by two main factors: the level of knowledge regarding BC risk factors, particularly obesity, and the familiarity with screening methods for early BC detection and improved prognosis [11]. Mammography plays a vital role in ensuring early diagnosis, increasing the chances of effective treatment and recovery [13]. The Centers for Disease Control and Prevention (CDC) recommends mammography for breast cancer screening at least once every two years for women aged 50–74 years [13]. In Saudi Arabia (KSA), this age group accounts for 5.1% of the country’s population of 20 million. While other screening methods, such as breast self-examinations and clinical breast examinations, were previously considered effective for detecting breast cancer, they are no longer recommended in updated breast cancer screening guidelines [12,13]. This study’s significance stems from the necessity of examining Saudi women’s awareness of BC risk factors, especially obesity, in the Qassim region. This study aimed to measure the knowledge and awareness level of Saudi women in the Qassim region regarding obesity as an associated risk factor for BC and examine their practice and awareness level of BC early screening and diagnostic methods, which help enable early BC detection and better prognosis. Moreover, the goal of the study was to increase the level of awareness regarding the impact of obesity and its relationship with the incidence of BC to gauge society’s awareness and knowledge levels about this connection, hoping to shed light on the understanding of this crucial health relationship in the Qassim region, Saudi Arabia.

## 2. Materials and Methods

### 2.1. Study Design and Duration

This observational, cross-sectional study was conducted from January 2024 to March 2024 to assess knowledge and awareness levels regarding the impact of obesity as a risk factor for breast cancer (BC). Additionally, the study measured participants’ awareness and practices regarding screening methods for early BC detection among Saudi women in the Qassim region.

### 2.2. Sample Size, Sampling Technique, and Study Location

The sample size was calculated using the standard sample size equation: n = z^2^ p (1 − p)/e^2^. Assuming the prevalence of good knowledge regarding breast cancer risk factors among Saudi females in the Qassim region as 50%, with a precision of 5% and a confidence interval of 95%, a sample size of 400 was accepted.

A stratified random sampling technique was followed in this study. From the total of 400 participants, 200 participants were recruited through an online platform; the responses of 150 of them were accepted but 50 respondents’ responses were discarded as they were incomplete and missed some answers. Then, 250 females were randomly recruited from community health centers in the Qassim region after gaining their informed consent, and face-to-face interviews were conducted to help participants complete the questionnaire.

### 2.3. Informed Consent

Written informed consent was obtained from all the participants after explaining the purpose and importance of the study. Participants who were recruited online provided consent via an electronic form (Appendix A). Participants were informed of their right to withdraw from the study at any time, and all collected data were kept confidential and used exclusively for research purposes, following the Helsinki Declaration guidelines.

Inclusion Criteria: Saudi women from the Qassim region aged 18 years and older who agreed to participate.

Exclusion Criteria: non-Saudi women, women under 18 years, participants who declined to participate, and those who resided outside the Qassim region.

### 2.4. Data Collection

The questionnaire was specifically designed to evaluate participants’ knowledge and awareness level of the relationship between obesity and BC. Data were collected from January 2024 to March 2024 through an online questionnaire by the Google app and then published on different social media in both Arabic and English formats. The questionnaire was validated and was previously published as part of the National Cancer Institute’s online BCRA or the Gail Risk Assessment Tool [22] (Appendix A). A pilot group of 50 participants was investigated on 2 January 2024 to validate the results of the questionnaire, and the decision was taken by the authors to verify the suitability of the questionnaire length and type of questions. The participants answered some closed-ended multiple-choice questions in 10–15 min. Those who could not complete the questionnaire due to any reason or samples with non-answered or incomplete questions were excluded from the study analysis.

The questionnaire was composed of 3 sections to cover the aim of the study.

The first section was for socio-demographic characteristics, the second section was for evaluation of the level of knowledge, and the third section was for assessment of the practice and awareness levels of the respondents towards BC and obesity relationships and BC screening methods.

A total of 22 questions were included in the survey as follows:**A.** **Socio-Demographic Characteristics:** This section collected data on age in years, marital status, education level, weight in Kg, body mass index (BMI), exercise pattern, healthy diet, menstrual stage, obesity inheritance in the family, and source of knowledge. BMI was calculated for each participant by applying the formula of (weight [kg])/(height [m])^2^ and was classified according to the WHO classifications. Height was reported by participants’ self-report, and weight was measured in the primary health office, as per clinic protocol. The 150 participants who were recruited through an online platform recorded their body weight themselves.**B.** **Knowledge Evaluation:** Questions assessed participants’ understanding of BMI calculation, the role of weight gain as a BC risk factor, and other risk factors such as hormonal drugs, hereditary causes, and the relationship between postmenopausal obesity and BC risk.**C.** **Practice and Awareness of Screening Methods:** This section evaluated participants’ awareness and practices regarding breast self-examination (BSE) and mammography. Questions included awareness of annual examinations, who should perform them, and satisfaction with community awareness about BC in the Qassim region.

A total score for each participant was calculated by summing the scores of their answers. In addition, the level of knowledge was calculated by summing the scores of all knowledge questions. Out of a maximum score of 100, the results were divided into three main categories: poor (25–49%), moderate (50–74%), and good (75% and above). All the detailed information is listed in a STROBE checklist in the Appendix A.

### 2.5. Institutional Review Board Ethical Approval

Ethical approval for this study was obtained from the Committee of Research Ethics, Deanship of Scientific Research, Qassim University (Approval No. 24-77-03). Participants were informed about the study objectives and assured of data confidentiality. Written consent was obtained before participation.

## 3. Statistical Data Analysis

Collected data were entered in Microsoft Excel 2016 and then analyzed by EPI INFO 7. Descriptive statistical analysis was used to analyze the socio-demographic characteristics of the participants. Distribution and numerical data are illustrated as means ± SD. Categorical data are shown as frequencies (percentage). The chi-square test was used to test the association between non-quantitative variables. To examine the most vital variables as indicators of good knowledge, practice, and awareness of BSE practice with obesity as a risk variable for BC, a logistic regression analysis was applied. All the variables were expected to be correlated with good knowledge and awareness. Knowledge and BSE practice and awareness were considered as the dependent variables. For the logistic regression model regarding BSE practice and awareness, knowledge levels related to breast cancer were added to the independent variables. A *p*-value of ≤0.05 was considered as the level of statistical significance. The results were interpreted in Microsoft Word 2016 in the form of tables and graphs.

## 4. Results

The current study comprised a total of 400 female respondents. Table 1 illustrates that the significant majority of participants fell within the age range of 18 to 25 years, comprising 59.75% of the total sample.

Notably, the majority of these individuals were single, accounting for 71.25% of the surveyed population. Furthermore, the weight distribution indicated that over half of the participants, specifically 57.75%, were within the weight range of 40 to 60 kg. BMI recorded 20 participants (5%) as underweight, 178 (44.5%) as normal, 157 (39.25%) as overweight, and 45 (11.25%) as obese.

Educationally, a substantial proportion of participants held a bachelor’s degree, constituting 76% of the surveyed cohort. A family history of breast cancer was recorded in 89 (22.25%) of the participants, while 311 (77.75%) had no family history of BC. A total of 225 (56.25%) of the participants never exercise and 147 (36.75%) of them exercise three times or more a week. Only 73 of the participants consumed healthy diets (18.25%), while the majority of them did not (81.75%). The results show that most of the participants do not follow a healthy diet (81.75%), and half of them do not exercise (56.25%). Most of the participants were at the menstruating stage, 327 (81.75%), while 23 (5.75%) of the participants were in the postmenopausal phase. Most of the participants do not have hereditary obesity in their family (81.75%), while a small percentage of participants have it (18.25%). Most of the participants obtained their knowledge of BC risk factors from the internet and social media (43.5%); 8.75% of them gained their information from reading and the lowest percentage was from TV (3.25%).

Table 2 and Figure 1 demonstrate the assessment of the participant’s knowledge about obesity and BC. Approximately half of the participants do not know how to calculate their estimated BMI (53.75%). A small percentage of the participants knew that using hormonal drugs was considered a risk factor for BC (7.75%), while most of the participants did not know the risk of hormonal drugs in BC (92.25%). Regarding the participants’ knowledge about weight-gain drugs and the risk of BC, around half of the participants thought that consumption of weight-gain drugs was a risk factor for breast cancer (n = 230, 57.5%), while the minority thought it had no effect (n = 170, 12.5%) (Table 2). The participants showed poor knowledge about obesity after menopause as a risk factor for breast cancer (n = 196, 49%). Additionally, over half of the participants, specifically 204 respondents (51.0%), did not consider overweight and obesity as risk factors for breast cancer, as demonstrated in Table 2. The total knowledge score of our respondents was poor, at 42%. We assessed the association between the level of knowledge and different factors (age, marital status, educational level, family history of BC, and source of knowledge) using X^2^ and adjusted odds ratio (ORs). In the multivariate logistic regression analysis, a statistically significant association was found between age (*p* = 0.024, Adj OR = 2.21), educational level (*p* = 0.042, Adj OR = 1.92), and source of knowledge (*p* = 0. 036, Adj OR = 1.71) and good knowledge. Family history of BC and marital status were not significantly associated with knowledge scores (*p* > 0.05).

Table 3 and Figure 2, Figure 3 and Figure 4 represent the level of awareness and practice of performing self-examinations of the breast; most participants were aware of the need to perform self-examinations of the breast at home (n = 338, 84.5%) (Table 3, Figure 2). Moreover, more than half of the participants showed good awareness about how to perform a self-examination of the breast at home (n = 238, 59.6%). However, most of them do not perform a monthly self-examination (n = 365, 91.3%) (Table 3, Figure 3).

Despite the high education level of the participants, the majority of them had never undergone a mammogram examination (n = 381, 95.25%), and 365 (91.25%) of them did not conduct monthly self-examinations, which reflects the very poor awareness among the participants towards BC screening methods (Table 3).

A total of 292 (73%) of the participants had been made aware of the need for breast examination. Of these, 239 (81.8%) of them had heard about the importance of an annual breast examination from their doctor, while a small percentage were made aware through other ways (18.2%); 26.9% of the participants had never been made aware of it. More than half of the participants (n = 231, 57.75%) thought there was insufficient awareness of breast cancer in Saudi Arabia, and only 169 (42.25%) of them were satisfied with the level of awareness of BC in Saudi Arabia (Table 3, Figure 4).

According to the multivariate logistic regression analysis with an Adj OR, there was no significant correlation between the tested factors (age, marital status, educational level, family history of BC, obesity, and their main source of knowledge) and performing BSE (*p* > 0.05), while completing a mammogram screening was significantly linked with educational level (*p* = 0.02; Adj OR = 2.5), source of knowledge (*p* = 0.04, Adj OR = 2.4), and good knowledge (*p* = 0.03, Adj OR = 2.23).

## 5. Discussion

Breast cancer is the most common cancer and the second leading cause of mortality among women in Saudi Arabia [23]. According to the Ministry of Health, a woman is diagnosed with breast cancer every three minutes [23]. In Saudi Arabia, approximately 73% of women seek medical consultation only at advanced stages of the disease, making treatment more challenging [24]. Globally, breast cancer accounts for 11.7% of all cancers, with approximately 2.3 million new cases reported in women in 2020 [24]. In the Middle East, the incidence of breast cancer is alarmingly high, likely due to delayed detection and poor prognosis [25].

This study aims to measure the knowledge and awareness levels of Saudi women regarding obesity as a risk factor for breast cancer and the methods of screening among women in the Qassim region. Additionally, it seeks to increase awareness about the impact of obesity on breast cancer and to assess society’s understanding of this critical health relationship.

A significant proportion (59.75%) of the participants were between 18 and 25 years old. Weight distribution data revealed that over half of the participants (57.75%) weighed between 40 and 60 kg. Most participants (81.75%) were menstruating, while 5.75% were postmenopausal. The majority (81.75%) did not have a family history of hereditary obesity, whereas 18.25% reported a family history of obesity.

When measuring the level of knowledge about BC risk factors, the results showed a lack of knowledge about the estimated BMI; 53.8% of them did not know how to calculate BMI as an indicator of obesity status, which reflects poor knowledge. A small percentage of the participants used hormonal drugs (7.7%), while most of the participants did not consume any hormonal drugs (92.3%); however, they did not know if hormonal drug consumption was a risk factor for BC.

Moderate knowledge was observed regarding weight-gain drugs and their association with breast cancer risk. Around 57.5% of participants believed that consuming weight-gain drugs could increase the risk of breast cancer, while 12.5% thought it had no effect. The participants demonstrated poor knowledge about the link between obesity after menopause and breast cancer risk, with only 49% recognizing the connection. Over half (51.0%) did not consider overweight and obesity as risk factors for breast cancer, despite scientific evidence supporting this link, especially among postmenopausal women [11,12,13,14]. Multivariate logistic regression analysis revealed a statistically significant association between age, educational level, and source of knowledge and good knowledge scores. However, family history of breast cancer and marital status were not significantly associated with knowledge scores (*p* > 0.05).

Additionally, the BMI at diagnosis was linked with variations in the BC molecular subtype distribution [15,16]. This association is thought to be linked to three hormonal systems, one involving insulin and insulin-like growth factors (IGFs), particularly IGF-binding protein-1 (IGF-1), another related to sex hormones, and the last involving adipokines [17]. Our results were consistent with other studies conducted in Hail [26], Jeddah [27], and Riyadh [28].

The level of concern about breast cancer varies across regions in Saudi Arabia. Studies in other countries reported that over 30% of women sought medical attention more than three months after the onset of breast cancer symptoms [29,30]. In a study by Almeshari et al., 96.9% of the participants knew that early detection of BC is essential for improving treatment outcomes [26].

Regarding the level of awareness and practice of breast self-examination, most participants were aware of the need to perform a BSE (84.5%). Moreover, more than half of the participants showed moderate awareness about how to perform a BSE at home (59.6%). However, most of them did not perform a monthly BSE (91.3%). This lack of BSE practice may be attributed to the complexity of the practice and may be influenced by many factors, such as age, type of job, level of BC knowledge, and BSE awareness. BSE is a simple, noninvasive, and cost-effective method crucial for early detection of breast cancer. Lack of BSE awareness means delayed diagnosis and frequently poor survival rates [26]. Almeshari et al. [26] found that 92.2% of participants in their study lacked knowledge of how to perform a BSE, attributing this to a lack of time, fear of discovering a lump, educational level, and insufficient BSE awareness. These results are in parallel with previous studies that tested BC knowledge, awareness, and BSE practices among women in Saudi Arabia [31,32,33].

Despite high education levels among participants in this study, the majority (95.25%) had never undergone a mammogram, reflecting a poor level of awareness. While 73% of participants were informed about breast examinations, 81.8% of them gained awareness of the importance of an annual breast examination from their doctors, 18.2% gained awareness through methods other than from their doctors, and 26.9% of the participants had never been made aware.

A study by Almeshari et al. revealed that women were also unaware that a mammogram would be performed and reported being fearful of the procedure [26]. The multivariate logistic regression analysis with Adj ORs indicated no significant correlation between age, marital status, educational level, family history of breast cancer, and obesity and performing BSE (*p* > 0.05). However, performing mammograms was significantly linked to educational level and knowledge source.

In parallel to our findings, a systematic review by Alrajhi et al. [34] reported inadequate BC awareness and BSE practices among Saudi women, with only 20.2% demonstrating high knowledge levels, 13.5% a moderate level of knowledge, and 66.3% a low level of knowledge, while 41% practiced BSE.

Almutairi et al. [35] reported opposite results in their study in Riyadh City. Their participants exhibited a high level of BC awareness. They explained that Riyadh is the capital of Saudi Arabia, with a huge healthcare and educational system. Moreover, the participants acknowledged family history as a risk factor for BC, which may have led women to take precautions and try to gain more knowledge through variable educational resources and early preventive measures [35].

In Abha City, a cross-sectional study of 1092 women from urban primary healthcare centers highlighted poor awareness of mammography (22.0%) and limited BSE practice (41.5%). Similarly, Radi’s study in Jeddah City on 200 women indicated that factors such as education, employment status, and marital status influenced breast cancer knowledge and BSE awareness [36]. Furthermore, a study by Radi in Jeddah City on 200 women revealed the impact of education, employment status, and marital status on BC knowledge and BSE awareness, putting these factors into consideration when arranging directed educational campaigns for BC awareness in the region [37].

Consistent with our results, Ashareef et al. [38] conducted a study on 400 female school teachers in Makkah, KSA, and they reported that the majority of participants had a poor level of knowledge about BC symptoms. Moreover, 40% of them had not practiced any breast examination [38].

Alsareii et al. [39] observed that while 75.3% of participants in Najran exhibited adequate general knowledge about breast cancer, 94.3% had poor knowledge of its BC warning signs.

The present study revealed that over half of the participants (57.75%) believed there is insufficient awareness of breast cancer in Saudi Arabia, with only 42.25% satisfied with the current level of awareness; more than half did not perceive overweight and obesity as risk factors for breast cancer.

A similar study conducted by Alduraibi reported insufficient knowledge and low BSE practice among female teachers in Buraidah City, with only 14.8% performing monthly BSE. Lack of information about SBE and fear of discovering a lump were the main causes for neglecting BSE. Additionally, 22.5% of the female teachers had never undergone a mammogram [40].

Awareness gaps regarding breast cancer risk factors extend beyond Saudi Arabia. A national study in the United Arab Emirates (UAE) found that only 7.6% of women had a high level, 65.8% had a moderate level, and 19% had a poor level of BC awareness [41]. These findings underscore the need for educational campaigns to improve knowledge and promote early detection practices, particularly in the Qassim region, Saudi Arabia.

Despite 76% of participants holding a bachelor’s degree, unhealthy lifestyle habits, such as poor diet (81.75%) and lack of exercise (56.25%), were prevalent. Nutritional and social habits are critical for health evaluation and intervention [42], and eating disorders are presented as an important public health issue [42]. Sandri et al. [43] developed and psychometrically tested a scale capable of assessing variable aspects of health globally, called the “Nutritional and Social Health Habits Scale” (NutSo-HH). They claimed that nutritional and social habits are crucial for better health evaluation that may permit the creation of specific individualized interventions to accelerate individual nutrition and health habits [43]. The NUTRI score enables an estimation of the effect of nutrition on health by calculating various food groups’ consumption frequency and their health consequences. The NUTRI score emphasizes that more protein-abundant food consumption (2–4 times per week) is highly recommended; in addition, daily fruit and vegetable consumption is recommended [44]. However, fast, fried, and processed foods and sugary soft drinks are not recommended and have multiple health drawbacks [45]. Unfortunately, a valid unified score for measuring variable aspects of the Saudi population’s health and different aspects of healthy habits is lacking. NutSo-HH and its ability to measure various aspects of healthy habits need to be validated in Saudi women in further studies to enable screening of nutritional and socio-economic habits.

## 6. Conclusions

The current findings strongly indicate a noticeable scarcity of knowledge and awareness regarding obesity-related BC risk among Saudi women in the Qassim region. The rising incidence of breast cancer in Saudi Arabia, coupled with limited knowledge of risk factors and early detection methods, necessitates urgent action in the Qassim region. Educational awareness campaigns using mass media, social media, and field awareness programs are essential to address this public health concern. Hopefully, our findings may help in encouraging women to perform regular BSE at home and undergo annual mammograms, which could improve early detection rates and reduce breast cancer mortality in the Qassim region, Saudi Arabia.

## 7. Limitations

This study has several limitations. The small sample size, few measured tested variables, and the cross-sectional nature of the study using a self-administered questionnaire, either face-to-face or online, may lead to unbiased answers. The study examined the knowledge and awareness of Saudi women aged 18 years and older only; information on younger females’ awareness is lacking in this study. The study did not examine the psychological effect on the body weight status and risk of BC.

## 8. Recommendation

This article’s importance originates from the necessity to measure the knowledge and awareness level of Saudi women regarding obesity and its association with breast cancer, as well as the practice of SBE screening among Saudi women in the Qassim region. Hopefully, our results will encourage further longitudinal studies with larger sample sizes and variable BC risk factors which will help to shed light on this crucial health relationship and improve the knowledge and awareness of Saudi women regarding obesity as a risk factor of BC, enabling the early detection and a better prognosis of the disease and hopefully saving many women’s lives.

## Figures and Tables

**Figure 1 healthcare-13-00278-f001:**
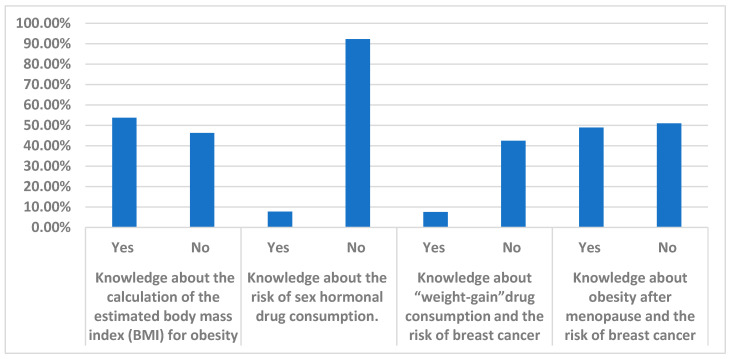
Percentages of participants’ knowledge regarding obesity as an associated risk of breast cancer; n = 400. BMI; body mass index.

**Figure 2 healthcare-13-00278-f002:**
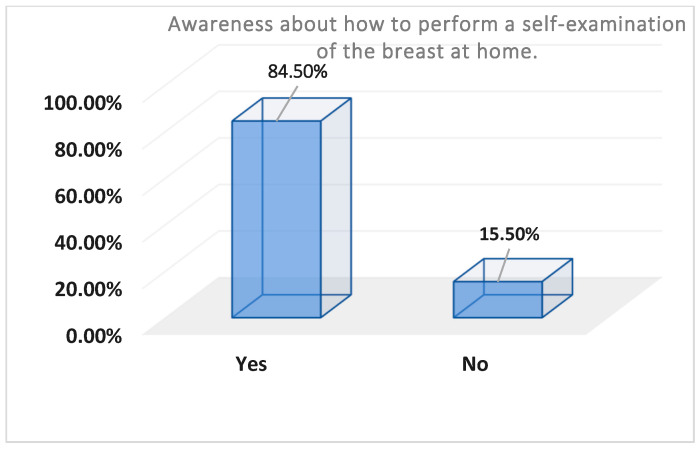
Percentages of participants’ awareness about the method of breast self-examination; n = 400.

**Figure 3 healthcare-13-00278-f003:**
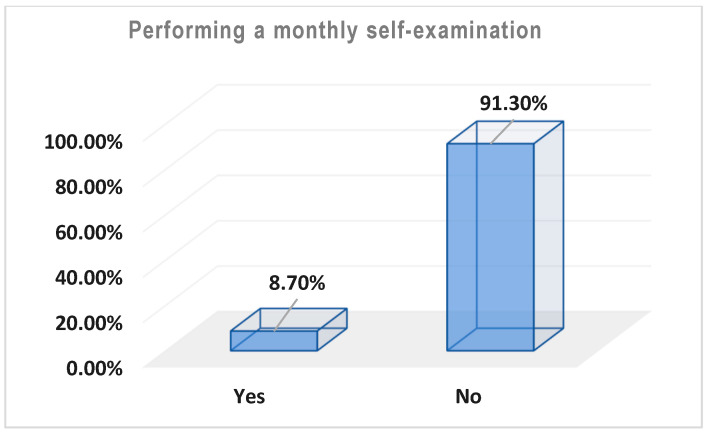
Percentages of the monthly performance of breast self-examination among the participants; n = 400.

**Figure 4 healthcare-13-00278-f004:**
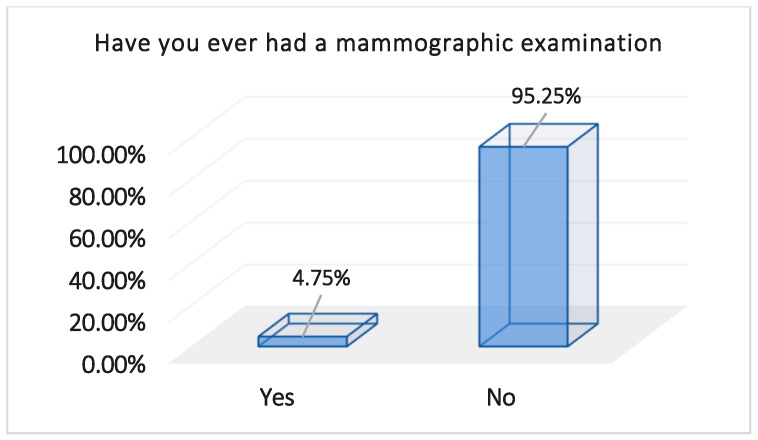
Percentages of mammograms performed by the participants; n = 400.

**Table 1 healthcare-13-00278-t001:** Description of the socio-demographic and lifestyle characteristics of the study sample.

	Demographic Characteristics		Total N = 400	
			N	(%)
1	Age (years)	18–25	239	59.75%
		26–39	119	29.75%
		>40	42	10.5%
2	Marital status	Married	115	28.75%
		Single	285	71.25%
3	Education level	Middle school	19	4.75%
		High school	65	16.25%
		Diploma	12	3.0%
		Bachelor’s	304	76.0%
4	Family history of breast cancer	Yes	89	22.25%
		No	311	77.75%
5	Body weight (Kgs)	40	15	3.75%
		40–60	231	57.75%
		61–80	115	28.75%
		81–100	31	7.75%
		>100	8	2.0%
6	Body mass index (BMI) kg/m^2^	Underweight (<18.5)	20	5%
		Normal (18.5–24.9)	178	44.5%
		Overweight (25.0–29.9)	157	39.25%
		Obese (>30)	45	11.25%
7	How often do you exercise?	Never	225	56.25%
		Once to twice a week	28	7%
		Three times or more a week	147	36.75%
8	Healthy dietary habits	Yes	73	18.25%
		No	327	81.75%
9	Menstruation status	Menstruation	327	81.75%
		Menopausal	50	12.5%
		Postmenopausal	23	5.75%
10	Hereditary obesity in the family	Yes	73	18.25%
		No	327	81.75%
11	Sources of knowledge regarding breast cancer	Media (TV)	13	3.25%
		Reading	35	8.75%
		Family/Friends	67	16.75%
		Educational lectures	111	27.75%
		Internet	174	43.5%

**Table 2 healthcare-13-00278-t002:** Numbers and percentages of participants with knowledge regarding obesity and breast cancer factors, with the *p*-value of the chi-square (X^2^) test.

The Participant’s Knowledge Regarding Obesity and Breast Cancer		Total N = 400 N (%)	*p*-Value
**1.** **Knowledge about the calculation of the estimated body mass index (BMI) for obesity**	Yes No	215 (3.75%) 185 (6.25%)	0.612 *
**2.** **Knowledge about the risk of sex hormonal drug consumption**	Yes No	31 (7.75%) 369 (92.25%)	0.001 **
**3. Knowledge about weight-gain drug consumption and the risk of breast cancer**	Yes No	230 (7.5%) 170 (42.5%)	0.042 **
**4. Knowledge about obesity after menopause and the risk of breast cancer**	Yes No	196 (49%) 204 (51%)	0.786 *

* Non-significant difference. ** Significant difference.

**Table 3 healthcare-13-00278-t003:** Participants’ practice and awareness of the need for an annual breast cancer examination and their information source with the *p*-value of the chi-square (X^2^) test.

Practice and Awareness of an Annual Breast Cancer Examination and Their Information Source	Yes N (%)	No N (%)	* p * -Value
**1. Are you performing the self-examination of the breast at home?**	338 (84.5%)	62 (15.5%)	*p* = 0.001
**2. Are you aware about how to perform a self-examination of the breast at home?**	238 (59.6%)	162 (40.4%)	*p* < 0.05
**3. Are you performing a monthly breast self-examination?**	35 (8.7%)	365 (91.3%)	*p* = 0.001
**4. Have you ever had a mammographic examination (mammogram)?**	19 (4.75%)	381 (95.25%)	*p* = 0.001
**5. Have you ever been made aware of the need for a breast examination?**	292 (73%)	108 (27%)	*p* = 0.001
**If yes, what is your source of information?**			
**Doctors**	239 (81.8%)		
**Other sources**	53 (18.2%)		
**Do you think there is enough awareness about breast cancer risk factors in Saudi Arabia?**	231 (57.75%)	169 (42.25%)	*p* < 0.05

## Data Availability

No new data were created or analyzed in this study. Data sharing is not applicable to this article.

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
