# Peer review of "Knowledge and Awareness of Obesity-Related Breast Cancer Risk Among Women in the Qassim Region, Saudi Arabia: A Cross-Sectional Study"

_healthcare, 2025, doi:10.3390/healthcare13030278_

Round 1
Reviewer 1 Report
Comments and Suggestions for Authors
Dear Authors,
the comments in the annex file
Best

Author Response
For research article
|
Response to Reviewer X Comments
|
||
|
1. Summary |
|
|
|
Thank you very much for taking the time to review this manuscript. Please find the detailed responses below and the revisions/corrections highlighted/in track changes in the re-submitted files..
|
||
|
2. Questions for General Evaluation |
Reviewer’s Evaluation |
Response and Revisions |
|
Does the introduction provide sufficient background and include all relevant references?
|
Yes/Can be improved/Must be improved/Not applicable |
Agree , it has een revised and modified and improved. |
|
Are all the cited references relevant to the research? |
Yes/Can be improved/Must be improved/Not applicable |
|
|
Is the research design appropriate? |
Yes/Can be improved/Must be improved/Not applicable |
The authors agree with the reviewer's comments and modified the research design to be clearer and more specific |
|
Are the methods adequately described? |
Yes/Can be improved/Must be improved/Not applicable |
The authors agree with the reviewer's comments and modified the research methodology to be more clear and specific.
|
|
Are the results clearly presented? |
Yes/Can be improved/Must be improved/Not applicable |
The authors modified the results to be more clear and indicative. |
|
Are the conclusions supported by the results? |
Yes/Can be improved/Must be improved/Not applicable |
Conclusion in the text and the abstract are modified to be conclusive, clear, and supported by the results , the conclusion modification is listed in the detailed point by point response as follows : |
|
3. Point-by-point response to Comments and Suggestions for Authors |
||
|
Comments : Are the conclusions supported by the results? Response: The conclusions of the manuscript appear to be supported by the results. The findings indicate a noticeable lack of knowledge and awareness among Saudi women in the Qassim region regarding obesity as a breast cancer risk factor. This is evidenced by the poor scores in the knowledge assessment (e.g., 51% of participants did not consider obesity a risk factor for breast cancer), low practice of breast self-examinations (91.3% did not perform monthly self-examinations), and extremely low mammogram usage (only 4.75% had undergone the examination). Additionally, the statistical analysis highlighted associations between factors like age, education, and sources of knowledge with awareness and practice levels, reinforcing the need for targeted educational interventions, which aligns with the study's recommendations for public health campaigns.
Comments 1: Editing: The legend for the acronyms used in the summary tables is missing, and the use of acronyms is inconsistent and poorly structured. Moreover, there are missing references to specific lines, which complicates the organization of the suggestions. The references are not formatted according to the journal’s template. . Response 1: Thank you for pointing this out, the authors agree and we added the full name of the abbreviation and acronyms listed in the tables and figures, the full name of some abbreviations are listed in the legends below Figure 1, while the remaining not needed as we wrote the full name in the tables cells. Moreover, missing references to specific lines have been added. Finally, the authors modified the references according to the journal’s template.
|
||
|
|
||
|
Comments 2: Title: I suggest making it clearer, possibly by shortening it but making it more specific in terms of the setting and population, e.g., Saudi Women… Saudi Arabia... |
||
|
Response 2: Agree. We have, accordingly, modified the title to emphasize this point. We briefed it and made it more specific as follows: Knowledge and Awareness of Obesity as an Associated Risk Factor for Breast Cancer among Saudi Women in The Qassim region, Saudi Arabia.
Comments 3 : Abstract: There is a lack of real interpretation in the conclusions regarding the implications for clinical practice of the studied phenomenon, with potential suggestions and specific proposals for the relevant scientific community. Response 3 : Agree. We have, accordingly, modified the conclusion and added some suggestive plans to increase the awareness and knowledge with encouragement the breast self-examination among Saudi women in the Qassim region and colored this addition in yellow as follows in the abstract : The findings highlight a noticeable gap in knowledge and awareness about obesity-related BC risk, as well as a limited awareness of breast self-examination and mammogram screening. These results underscore the urgent need for targeted awareness campaigns and educational programs in the Qassim region to address this critical health issue. Promoting BSE practices, weight management, and regular mammogram screenings could significantly enhance early detection, improve prognosis, and reduce BC-related mortality among Saudi women. The conclusion section in the text was as follows The current findings strongly indicate a noticeable scarcity of knowledge and awareness regarding obesity-related BC risk among Saudi women in the Qassim region. The rising incidence of breast cancer in Saudi Arabia, coupled with limited knowledge of risk factors and early detection methods, necessitates urgent action in the Qassim region. Educational awareness campaigns using mass media, social media, and field awareness programs are essential to address this public health concern. Hopefully, our findings may help in encouraging women to perform regular BSE at home and undergo annual mammograms could improve early detection rates and reduce breast cancer mortality in the Qassim region, Saudi Arabia.
Comments 4 : Keywords: I recommend using four or five keywords, more focused on the main topics of the study. The current ones are too sensitive. Response: Agree. We have, accordingly, modified the Key words to be more specific as follows: Breast Cancer, Risk Factors, Obesity, Knowledge, Awareness, Breast self-examination, Women's Health Screening.
Comments 5: Objectives: They are structured in a rather unclear and confusing way. I suggest being more specific. Response 5: The authors agree with the reviewers comments and modified the objectives to be as follows: The study aims to measure the knowledge and awareness levels of Saudi women in the Qassim region regarding obesity as a BC risk factor. Additionally, it seeks to evaluate their practices and awareness regarding early BC screening and diagnostic methods, which are crucial for early detection and better prognosis. Moreover, the study aims to raise awareness about the impact of obesity on BC incidence, ultimately improving and shedding light on the societal understanding of this critical health relationship in the Qassim region, of Saudi Arabia.
|
||
|
Comments 6: Methods: This is the most controversial aspect and certainly requires more attention, namely the lack of a structured reporting method, such as the STROBE checklist, which could have been used to ensure the scientific validity and transparency of the study. Including this checklist as a supplementary file and citing it in the text would be crucial to improve the quality and reproducibility of the study, making it more transparent and scientifically valid.
Response 6:Authors agreed with the reviewer's comment and prepared a STROBE checklist with detailed information and attached it as appendix 3 in the supplementary file attachment. The methodology section of the study included details about the study design and the periods of recruitment.2.2 Sample size, sampling technique, and study location. We clarified the method of determining the sample size and the equation used. All the details of the participants are included in this section, including the study location in healthcare centers in the Qassim region. Informed consent, the inclusion and exclusion criteria. Data collection details and the survey questionnaire validation were included as follows The questionnaire was validated and was previously published using the National Cancer Institute’s online BCRA or the Gail Risk Assessment Tool [22] ( Appendix 2 in the supplementary file ) . A pilot group of 50 participants was held on 2nd Jan 2024, to validate the results of the questionnaire, and the decision was taken by the authors to verify the suitability of the questionnaire length and type of questions appropriate. The participants answered some end-closed Multiple-Choice Questions, in 10-15 minutes. Those who could not complete the questionnaire due to any reason or samples with non-answered incomplete questions were excluded from the study analysis. All the survey Questionnaire details were presented clearly in the data collection section and the form of the questionnaire used was attached as supplementary files All the modified data are highlighted in yellow color. Comments 7: Results: Clear and well-reported. Response 7: The authors appreciate the reviewers’ comments regarding the study results.
Comments 8: Discussion: I suggest incorporating a comparison with other care settings to support and enhance the proposed discussion. Additionally, I recommend expanding the discussion, possibly concluding with a proposal to extend research on specific tools for nutritional and social health assessment (e.g., Development and psychometric testing of the nutritional and social health habits scale (NutSo-HH): A methodological review of existing tools, doi: 10.1016/j.mex.2024.102768 / A systematic review of tools designed for teacher proxy-report of children's physical literacy or constituting elements, doi: 10.1186/s12966-021-01162-3) as a potential operational tool for managing nutritional screening, even in breast cancer patients. Response 8: The authors agree with the reviewer and modified the discussion according to the comment as follows : Eating disorders are presented as an important public health issue [42]. Sandri et al [43] developed and psychometrically tested a scale capable of assessing variable aspects of health globally called ``Nutritional and Social Health Habits Scale'' (NutSo-HH), they claimed that nutritional and social habits are crucial for better health evaluation that may permit the creation of specific individualized interventions to accelerate individual nutrition and health habits [43]. The factor NUTRI enables the score estimation of the effect of nutrition on health by calculating various food groups' consumption frequency and their health consequences, as NUTRI emphasizes that more protein-abundant food consumption (2–4 per week) is highly recommended, daily fruits and vegetable consumption is recommended [44]. However, fast, fried, processed foods and sugary soft drinks are not recommended and have multiple health drawbacks [45]. Unfortunately, a valid unified score for measuring variable aspects of the Saudi population's health and different aspects of healthy habits is lacking. NutSo-HH and its ability to measure various aspects of healthy habits need to be validated in Saudi women in further studies for nutritional and socio-economic habits screening. Comment 9: Limitations: In my opinion, these should be expanded. Response 9: The authors expanded this section as follows : -Limitations The study has several limitations. The small sample size, few measured tested variables, and the cross-sectional nature of the study using a self-administered questionnaire either face-to-face or online survey may lead to unbiased answers. with its impacts is another limitation. The study examined the knowledge and awareness of Saudi women aged 18 years and older only, the younger age females' awareness is lacking in this study. The study did not examine the psychological effect on the body weight status and risk of BC. 4. Response to Comments on the Quality of English Language |
||
|
Point 1: |
||
|
Response 1: Thank you for pointing this out, the manuscript has been edited to improve the English grammar and writing |
||
|
5. Additional clarifications |
||
|
The authors appreciate this valuable review that contribute to improve the manuscript scientific level and hope to be finally approved and accepted for publication in this highly ranked Journal Regards |
||
For review article
|
Response to Reviewer X Comments
|
||
|
1. Summary |
|
|
|
Thank you very much for taking the time to review this manuscript. Please find the detailed responses below and the corresponding revisions/corrections highlighted/in track changes in the re-submitted files. [This is only a recommended summary. Please feel free to adjust it. We do suggest maintaining a neutral tone and thanking the reviewers for their contribution although the comments may be negative or off-target. If you disagree with the reviewer's comments please include any concerns you may have in the letter to the Academic Editor.]
|
||
|
2. Questions for General Evaluation |
Reviewer’s Evaluation |
Response and Revisions |
|
Is the work a significant contribution to the field? |
|
This work of significant value as it discussed major health problem among Saudi women regarding BC and obesity. |
|
Is the work well organized and comprehensively described? |
|
The authors modified the article to be more comprehensive.
|
|
Is the work scientifically sound and not misleading? |
|
|
|
Are there appropriate and adequate references to related and previous work? |
|
The authors have added some relevant references of previous work as recommended.
|
|
Is the English used correct and readable? |
|
English editing has been done . |
Reviewer 2 Report
Comments and Suggestions for Authors
Introduction-
A thorough review by an English fluent editor is recommended.
Watch out for incomplete sentences: “Currently ranks…
Need to say BC currently ranks…
Write out WHO before you use the abbreviation
You need a citation for the sentence starting with: “The rate of breast cancer…”
In this sentence: “associated with cancer bad prognosis with a high mortality rate”
Change to “ is associated with poor cancer prognoses and higher mortality rates”
In the sentence: “accumulation of pro-inflammatory T cells and decreasing in T regs…”
Change to: “accumulation of pro-inflammatory T cells and a decrease in T regs, which is relevant to obesity-related insulin resistance”
From this sentence: “Moreover, to increase…”
change to “moreover the goal of this study was to increase…”
In the introduction the authors should talk about the availability of mammograms in the country and whether physicians perform clinical breast exams and discuss self exams during visits.
In the Sample Size paragraph, this sentence: “A stratified random… is confusing and awkward. Also why were some recruited in person and some online?
The paragraph on the survey tool is confusing – how was in used related to BRCA?
What does this mean: “audience awareness”?
The authors didn’t sum the number of the answers to get a total – they would have summed the scores for the answers. But what kind of scale was used?
Why when looking at BC screening would you choose such a relatively young sample? At what age does screening start in Saudi Arabia?
How was healthy diet defined?
What are these: “weight gain drugs”?
I don’t think the charts are needed – the tables are good enough
You don’t need to repeat BC statistics in the discussion – first 2 paragraphs are too much. One sentence about the purpose would be enough.
Don’t start sentences with a number.
Do not repeat the results in the discussion section.
The discussion section needs a rewrite
Do the authors think that an awareness of the connection between obesity and BC will discourage an unhealthy diet in women in KSA?
Author Response
Comments and Suggestions for Authors
Introduction-
Comment 1: A thorough review by an English-fluent editor is recommended.
Response1 . The authors agreed and did thorough English editing.
Comment 2: Watch out for incomplete sentences: “Currently ranks…
Need to say BC currently ranks…
Response 2: The authors considered this comment and modified according to this comment. As follows
BC currently ranks as the fifth highest leading cause of cancer-related deaths globally and in Saudi Arabia [1-2].
Comment 3 : Write out WHO before you use the abbreviation
Response 3 : The authors modified it as follows : World Health Organization (WHO)
Comment 4: You need a citation for the sentence starting with: “The rate of breast cancer…”
Response 4: The reference has been added. The rate of breast cancer according to the World Health Organization (WHO) has risen over the last two decades, from an evaluated 10 million in 2000 to 19.3 million in 2020 [3].
Comment 5: In this sentence: “associated with cancer bad prognosis with a high mortality rate”
Change to “ is associated with poor cancer prognoses and higher mortality rates”
Response 5 : Agreed and modified as ( Obesity has been detected to be linked with BC incidence and is associated with poor cancer prognoses and higher mortality rates [11,12].
Comment 6 : In the sentence: “accumulation of pro-inflammatory T cells and decreasing in T regs…”
Change to: “accumulation of pro-inflammatory T cells and a decrease in T regs, which is relevant to obesity-related insulin resistance”
Response 6: It has been modified as recommended.
Comment 7: From this sentence: “Moreover, to increase…”
change to “moreover the goal of this study was to increase…”
Response 7 : Thank you for pointing this point out, the authors modified it as required.
Comment 8 : In the introduction the authors should talk about the availability of mammograms in the country and whether physicians perform clinical breast exams and discuss self-exams during visits.
Response 8 : this portion has been added in the introduction as follows :
Mammography plays a vital role in ensuring early diagnosis, increasing the chances of effective treatment and recovery.[12] The Centers for Disease Control and Prevention (CDC) recommend mammography for breast cancer screening at least once every two years for women aged 50–74 years[12]. In Saudi Arabia (KSA), this age group accounts for 5.1% of the country’s 20 million population. While other screening methods, such as breast self-examinations and clinical breast examinations, were previously considered effective for detecting breast cancer, they are no longer recommended in updated breast cancer screening guidelines[12].
Comment 9: In the Sample Size paragraph, this sentence: “A stratified random… is confusing and awkward. Also, why were some recruited in person and some online?
Response 9: Stratified random sampling is a type of selecting the participants in which the population could be divided into smaller subgroups, or strata, based on shared characteristics of the members and then randomly selected among each stratum to form the final sample.
These shared characteristics can include age, sex, nationality, education level, or income, etc.
In the current study Only Saudi female participants from Al Qassim region grouped from 18 years or older are allowed to participate in this study.
400 participants; 200 participants were recruited through an online platform, responses of 150 of them were accepted but 50 respondents' responses were discarded as they were incomplete and missed some answers. 250 females were randomly collected from the community health centers in the Qassim region after gaining their informed consent, and face-to-face interviews were conducted to assist participants in completing the questionnaire.
Comment 10-: The paragraph on the survey tool is confusing – how was in used related to BRCA?
Response 10: The survey Questionnaire was attached as appendix 2 in the supplementary files.
Comment 11 What does this mean: “audience awareness”?
Response 11: Mean societal awareness in the Qassim region among the women lived Al Qassim region Saudi Arabia
The authors didn’t sum the number of the answers to get a total – they would have summed the scores for the answers. But what kind of scale was used?
Response 11: we made this modification in the data collection section as follows :
A total of 22 questions were included in the survey as follows:
A- Socio-Demographic Characteristics: This section collected data on age in years, marital status, education level, weight in Kg, body mass index (BMI), exercise pattern, healthy diet, menstrual stage, obesity inheritance in the family, and source of knowledge). BMI was calculated for each participant by applying the formula of (weight [kg])/(height [m])2 and was classified according to the WHO classifications. Height was reported by participants' self-report, and weight was measured in the primary health office, per clinic protocol. The 150 participants who were recruited through an online platform, recorded their body weight by themselves.
B-Knowledge Evaluation: Questions assessed participants' understanding of BMI calculation, the role of weight gain as a BC risk factor, and other risk factors such as hormonal drugs, hereditary causes, and the relationship between postmenopausal obesity and BC risk.
C-Practice and Awareness of Screening Methods: This section evaluated participants' awareness and practices regarding breast self-examination (BSE) and mammography. Questions included awareness of annual examinations, who should perform them, and satisfaction with community awareness about BC in the Qassim region.
- A total score for each participant was calculated by summing the number of their answers. In addition, the level of knowledge was calculated by summing the scores of all knowledge questions. Out of a maximum score of 100, the results were divided into three main categories, Poor (25–49%), Moderate (50–74%), and Good (75% and above).
Comment 12: Why when looking at BC screening would you choose such a relatively young sample? At what age does screening start in Saudi Arabia?
Response 12: The Centers for Disease Control and Prevention (CDC) recommend mammography for breast cancer screening at least once every two years for women aged 50–74 years [13]. In Saudi Arabia (KSA), this age group accounts for 5.1% of the country’s 20 million population. While other screening methods, such as breast self-examinations and clinical breast examinations, were previously considered effective for detecting breast cancer, they are no longer recommended in updated breast cancer screening guidelines [12, 13].
The authors do not exactly the statistics a what age in the Qassim region the females seek for the screening but it has been presented that the awareness of early screening methods is too poor and needs to be improved by several educational campaigns, health care centers efforts, social media and different tools to improve Saudi females knowledge and awareness regarding BC and its screening methods. That is the goal of our study.
Comment 13: How was a healthy diet defined?
Response 13: The factor NUTRI enables the score estimation of the effect of nutrition on health by calculating various food groups' consumption frequency and their health consequences, as NUTRI emphasizes that more protein-abundant food consumption (2–4 per week) is highly recommended, daily fruits and vegetable consumption is recommended [44]. However, fast, fried, processed foods and sugary soft drinks are not recommended and have multiple health drawbacks [45]
So they were asked if their meals consisted of high fruits, protein, nuts and good fats with low carbohydrates, sugary and fast food is considered healthy diet habits.
Comment 14 : What are these: “weight gain drugs”?
Response 14 : Any drugs with side effects of weight gaining, such as hormonal supplementation , contraceptive pills , or insulin hormone .
Comment 15 : I don’t think the charts are needed – the tables are good enough.
Response 15: The authors made these figures just for more clarification and statistical presentation
Comment 16: You don’t need to repeat BC statistics in the discussion – the first 2 paragraphs are too much. One sentence about the purpose would be enough.
Response 16: The authors agree but regarding these statistics for the BC in Saudi Arabia specifically, the authors preferred to keep them to show the importance of the study among Saudi women.
Comment 17: Don’t start sentences with a number.
Response 17: The authors agreed and modified the discussion section according to this comment.
Comment 18 : Do not repeat the results in the discussion section.
Response 18: The numbers in the discussion are just for clarification and comparison with other studies in different regions in Saudi Arabia and we made some abstracting and cutting as much as we could to keep the discussion comprehensive and conclusive.
Comment 19: The discussion section needs a rewrite
Response 19: This section has been modified and rewritten with more English editing process
Comment 20: Do the authors think that an awareness of the connection between obesity and BC will discourage an unhealthy diet in women in KSA?
Response 20: The authors consider this global health concern mainly among women in Saudi Arabia being
The current findings strongly indicate a noticeable scarcity of knowledge and awareness regarding obesity-related BC risk among Saudi women in the Qassim region. This necessitates urgent action in the Qassim region. Educational awareness campaigns using mass media, social media, and field awareness programs are essential to address this public health concern. Hopefully, our findings may help in encouraging women to perform regular BSE at home and undergo annual mammograms could improve early detection rates and reduce breast cancer mortality in the Qassim region, Saudi Arabia.
Round 2
Reviewer 1 Report
Comments and Suggestions for Authors
Dear Authors,
the comments in the annex file.
Best.

Author Response
|
3. Point-by-point response to Comments and Suggestions for Authors |
|
Comments 1: Title: “Saudi Women”, It refers to “Saudi Arabia” it is a repetition that it is best left as Women because it is later specified that is referred to ( Saudi Arabia).
|
|
Response 1: Thank you for pointing this out. I/We agree with this comment. Therefore, we have modified the title to Knowledge and Awareness of Obesity-Related Breast Cancer Risk Among Women in the Qassim Region, Saudi Arabia. A Cross-Sectional Study.
|
|
Comments 2: Methods: please cite in the text the Strobe Method applied for reporting and include the reference. This approach would scientifically support the manuscript later. |
|
Response 2: Agree. We have, accordingly, modified to emphasize this point. The Strobe checklist has been cited in the text in the method section as a supplementary file, Appendix 3. And colored in yellow to be marked , as follows All the detailed information is listed in a STROBE checklist in the supplementary file (Appendix 3 in the supplementary file) |
|
Comments 3 : Attention: Table 1 and lines 270-272 aren’t edited correctly. Response 3 : Agree. Table 1 has been edited and lines 270-272 have been modified as follows Breast cancer is the most common cancer and the second leading cause of mortality among women in Saudi Arabia [23]. According to the Ministry of Health, a woman is diagnosed with breast cancer every three minutes [23]. In Saudi Arabia, approximately 73% of women seek medical consultation only at thadvanced stages of the disease, making treatment more challenging.[24].
|
